


**Characteristics of Heatwaves in Africa: Morocco 2000 and South Africa 2015/16**
Chloe Brimicombe[1*]; Claudia Di Napoli[2], Rosalind Cornforth[3], Florian Pappenberger[4], Celia
Petty[3, 5] and Hannah L. Cloke[1, 6, 7, 8];
[1] Department of Geography and Environmental Science, University of Reading, RG6 6AB, UK.
[2] School of Agriculture, Policy and Development, University of Reading RG6 6AR, UK.
[3] Walker Institute, University of Reading, Reading, RG6 6AR, UK.
[4] European Centre for Medium-Range Weather Forecasts (ECMWF), Shinfield Park, Reading,
RG2 9AX, UK
[5] Evidence for Development, University of Reading, Reading, RG6 6AR, UK.
[6] Department of Meteorology, University of Reading, Reading RG6 6UR, UK
[7] Department of Earth Sciences, Uppsala University, SE-751 05 Uppsala, Sweden
[8] Centre of Natural Hazards and Disaster Science, CNDS, SE-751 05 Uppsala, Sweden
* Corresponding author: email c.r.brimicombe@pgr.reading.ac.uk
**Abstract:**
Heatwaves pose an ever increasing risk to African communities as exposure to heat extremes
can have a drastic effect on individuals and in some cases can even result in death. This study
presents new information about the characteristics of historical African heatwaves including
a comprehensive synopsis of documented heatwave events from 1980 until 2020.Detailed
research on heatwave case studies helps to inform the development of early warning systems
and forecasting, which is an urgent priority. Here, the focus is on two reported heatwaves,
Morocco 2000 and South Africa 2015/16. Both heatwaves feature in the EM-DAT database
and include reported impacts, with the Morocco heat being the only hazard to be associated
with an economic cost. In addition, these heatwaves reveal how the mechanisms behind them
are closely influenced by synoptic systems and geography of their regions. Further, It is
demonstrated there is some reporting by African Nations for heatwaves but that this needs
significant improvement.


## 1. Introduction

Africa is witnessing a rapid increase in dangerous weather extremes, including floods, drought and extreme heat (IPCC, 2013, Russo et al 2016). On a global scale heat extremes are known to be lethal, with more than 70,000 and 55,000 people dying globally in 2003 and 2010 (Schubert *et al* 2011, Robine *et al* 2008). However, there is surprisingly little evidence on heat extremes and their impacts across the African continent (Harrington and Otto, 2020; van der Walt and Fitchett , 2021).

Robust evidence is important for Africa for many reasons. It is a continent that is home to a rapidly increasing population, where these events represent a serious threat to local communities, and where many lack access to the resources needed to build resilience and adaptive capacity (Adger *et al* 2009, Russo *et al* 2016). The sparse evidence that exists demonstrates that heat in African nations can be deadly (Frimpong *et al* 2017, Codjoe *et al* 2020) and has many impacts for example reducing crop yields (Mubiru *et al* 2018, Mwaura and Okoboi 2014, Epule *et al* 2018, Abdulai *et al* 2018) and changing how communities migrate (Abass *et al* 2018, Gray and Wise 2016). Climate change projections show that heat could become so extreme in some regions of Africa that it will be uninhabitable unless there is urgent mitigation (Russo *et al* 2016, Rohat *et al* 2019, Schwingshackle *et at* 2021).

In addition there are calls by many for the international community to come to a consensus and face the risk posed by heat globally and in Africa (Harrington and Otto 2020, Russo *et al* 2016, Global Commission on Adaptation 2020, World Meteorological Organization 2018). This is a mandate for many working in the humanitarian sector who are on the frontline of climate emergency: *"An understanding of how temperatures have impacted people in Africa during past extreme heat events is critical to building public consensus on the issue and moving us toward action."* (Roop Singh, Climate Risk Adviser for the Red Cross Climate Centre, pers comm).

This study focuses on the extent to which heatwaves are recorded for the continent of Africa. It presents the first list of reported heatwaves in literature for the continent. Further, two heatwave case studies for Morocco July/August 2000 and South Africa 2015/16 and their characteristics are explored. This study sets a mandate for other reported heatwaves to be explored further and calls for more robust heat hazard reporting in African Nations.


## 2. Methods

In this study a mixed method approach is used. Analysis of literature is employed to provide a synopsis of reported heatwaves, before the focus shifts to meteorological analysis of two heatwave case studies of Morocco 2000 and South Africa 2015/16. This approach was chosen as it allows for a more useful analysis of case studies directly from literature, than reviewing literature alone.

### 2.1 Synopsis of Africa historic heatwaves

A synopsis of historic heatwaves that occurred in Africa between 1980 and 2020 to date was compiled based on analysis of 21 academic papers, grey literature and EM-DAT. The chosen literature had to meet the criteria of having a focus on historic heatwaves in Africa and include either a physical characteristic of the heatwave or impact. Grey literature includes the World Meteorological Organization (WMO) Reports (e.g. World Meteorological Organisation, 2013) and the American Meteorological Society (AMS) State of the Climate publications (e.g. American Meteorological Society, 2004).

### 2.2 Defining a Heatwave

There is not a universal definition of heatwaves, as such we define a heatwave using an adaptation of the World Meteorological Organisation (WMO) definition for the Universal Thermal Climate Index (UTCI). According to the WMO heatwaves can be defined as *"A period of marked unusual hot weather (maximum, minimum and daily average temperature) over a region persisting at least three consecutive days during the warm period of the year based on local (station-based) climatological conditions, with thermal conditions recorded above given thresholds"* (World Meteorological Organization 2018a). Following that we define a heatwave in terms of heat stress when the mean UTCI is above the climate 90th percentile for the region. This is above 26.1°C UTCI (the 90th Percentile for December and January 1981 to 2019) for South Africa and 28.2°C UTCI (the 90th Percentile for July and August 1981 to 2010) for Morocco.

The UTCI is a thermal index which makes use of the meteorological parameters of 2m temperature, 2m dew point temperature, 10m wind speed and mean radiant temperature and a body model (Di Napoli *et al* 2021). It has been compared to many other thermal indices



such as apparent temperature and heat index and captures well an average body response to
the thermal environment (Zare *et al* 2018, Jendritzky *et al* 2012, Blazejczyk *et al* 2012). It has
further been shown to be able to forecast heatwaves internationally (Pappenberger *et al*
2015) and accurately indicate extreme heat for Africa (Guigma *et al* 2020).
**2.3 Historical Heatwaves Anomalies**
This study provides analysis of anomalies of mean values of variables that are indicative of
heatwaves (Guigma *et al* 2020, Oueslati *et al* 2017) for 2m air temperature, the UTCI and
geopotential height at 500hpa (z500) and 850hpa (z850) for two case studies: the 2015/2016
South Africa heatwave and the 2000 Morocco heatwave from the ERA5 reanalysis data set
(Hersbach *et al* 2020; Di Napoli *et al* 2021). These heatwaves were chosen because they occur
in different climate regions of the continent and exhibit different heat characteristics, notably
area. In addition, they have reported impacts which are included in the EM-DAT international
disaster database (CRED 2020).
Anomalies were calculated by using the mean of the period of the heatwave, minus the
climatological values (1980 to 2010) of the months the heatwave is in, namely December and
January for the 2015/2016 South Africa heatwave, and July and August for the 2000 Morocco
heatwave. In addition, anomalies for the week before and the week after the heatwave are
calculated to provide a picture of the meteorological characteristics that start and end the
two heatwaves.  All calculations were carried out using Rstudio.











## 3. Results

### 3.1 Heatwaves are reported almost annually since 1980

Academic Literature and reports for Africa show that 39 heatwaves are reported for somewhere in Africa almost every year since 1980 (*Table 1*). Years where no reports could be found are 1984, 1985, 1991, 1993, 1994, 1991, 2001, 2006, 2007 and 2014. The synopsis also demonstrates where heatwaves are most commonly recorded. Specifically, it is seen that the Sahara region (17W, 36E, 17.5N, 30N) has the most reported heatwaves appearing 20 times whilst the Horn of Africa (39E, 50E, 5N, 12.5N) and Madagascar (30E, 50E, 30S, 10S) only appear once.

Characteristics of heatwaves are reported more than their impacts featuring for 26 of the heatwaves in comparison to 9 heatwaves respectively. The most reported characteristic for heatwaves is record temperatures (13), concurrence with El Niño-Southern Oscillation (ENSO) (12) and concurrence with drought (5). The most reported impact of heatwaves is mortality (6); other impacts include changes in social practices and agricultural impacts.


*Table 1: Synopsis of historic heatwaves compiled from news reports, academic literature and*
*climate reports for the African continent from 1980 to 2020.*

| Year [Season] | Region (corresponding with Table 2) | Heatwave Impacts | Heatwave Characteristics | References |
|---|---|---|---|---|
| 1982/3 | Sahel, Sahel-Sudan and Guinea | | Concurrent with the Sahel drought and an ENSO event | (Russo *et al* 2016) |
| 1987 [MAM] | South Africa | | Concurrent with a drought and an ENSO event | (Russo *et al* 2016) |
| 1988 | Sahara | | | (Russo *et al* 2016) |
| 1989 [SON] | Sahel, Sahel-Sudan and Guinea | | Locals reportedly remember high temperatures in Ghana | (Codjoe, *et al.*, 2020) |
| 1990 [SON] | Mediterranean | | | (Russo *et al* 2016) |
| 1992 | Mediterranean, Sahara, Congo and South Africa | | Concurrent with a drought | (Barbier *et al* 2018, Harrington and Otto 2020) |
| 1995 | Mediterranean and Sahara | 32 reported deaths in Egypt | | (CRED 2020) |
| 1996 | Mediterranean and Sahara | 22 reported deaths in Egypt | | (CRED 2020) |
| 1997/98 | Sahel-Sudan and Congo | | Concurrent with an ENSO event | (Oueslati *et al* 2017, Ceccherini *et al* 2017, Russo *et al* 2016) |
| 2000 [JJA] | Mediterranean and Sahara | News reports of 4 million chickens dying. Up to 809,000 USD of damage recorded in Morocco | | (Panafrican News Agency 2000, CRED 2020) |
| 2002 [JJA] | Sahara and Congo | 60 reported deaths in Nigeria | In the Sahara temperatures as high as 50.6°C during June and July 2002 | (World Meteorological Organisation 2013, CRED 2020) |
| 2003 [JJA] | Mediterranean and Sahara | 40 reported deaths in Algeria | Heatwave in Europe in the same season | (American Meteorological Society 2004, World Meteorological Organisation 2013, CRED 2020) |
| 2004[JJA] | Mediterranean, Sahara, Sahel, Sahel-Sudan and Congo | | ENSO event. Maximum temperature recorded on 29 June | (American Meteorological Society 2005, Russo |


| | | | in Sidi-Slimane, Morocco of 47°C. | et al 2016, Oueslati et al 2017) |
|---|---|---|---|---|
| 2004/5 | Mediterranean, Sahara, Sahel, Sahel-Sudan and Congo | | ENSO event | (Russo et al 2016) |
| 2005 [JJA] | Mediterranean and Sahara | | ENSO event | (World Meteorological Organisation 2013) |
| 2008 | Sahara and Congo | | | (Russo et al 2016) |
| 2009/10 | Sahara, Sahel, Sahel-Sudano and Guinea. | | ENSO event | (Russo et al 2016) |
| 2010 [JJA] | Mediterranean and Sahara but Morocco notable | | Heatwave in Europe in the same season and an ENSO event. | (Fontaine et al 2013, Russo et al 2016, World Meteorological Organisation 2013, Oueslati et al 2017) |
| 2011 | Congo, Mid Africa and Madagascar | | | (Russo et al 2016) |
| 2012 | South Africa | | | (Russo et al 2016) |
| 2012 [JJA, OND] | Mediterranean, Sahara, Sahel, Sahel-Sudano and Guinea | | Ouarglon, southern Algeria reports 50°C on 2nd August. In October Morocco records temperatures up to 36°C | (Achberger et al 2013, Russo et al 2016, Ceccherini et al 2017) |
| 2013 [MAM] | Sahara, Sahel, Sahel-Sudano, Guinea and South Africa. | | Warmest temperature up until 2013 recorded on the 6th March of 43.0°C in Navrongo, Ghana. In South Africa the hottest African temperature up until 2013 is recorded of 47.3°C on 4 March. Temperatures above 40°C recorded in Nigeria | (World Meteorological Organization 2015, Oueslati et al 2017) |
| 2013 [JJA] | Mediterranean and Sahara | | Daily maximum temperature anomaly of over 10°C recorded. | (World Meteorological Organization 2015) |



| 2015 [JJA] | Mediterranean and Sahara | | Heatwave reported to be lasted up to 40 days | (Benzerga 2015) |
|---|---|---|---|---|
| 2015 [MAM] | Mediterranean, Sahara, Sahel, Sahel-Sudano | | | (Russo *et al* 2016) |
| 2015 [JJA] | Mediterranean, Sahara, Sahel, Sahel-Sudano Congo, Horn of Africa | 110 death and 66 injuries reported in Egypt. 16 deaths reported in Sudan. | | (Hafez and Almazroui 2016, Russo *et al* 2016, CRED 2020) |
| 2015 [OND] | South Africa | | Concurrent with a drought and an ENSO event | (World Meteorological Organization 2016, Russo *et al* 2016) |
| 2016 [DJF] | Sahel, Sahel-Sudano, Guinea and South Africa | 11 deaths and 20 injuries reported in South Africa | Concurrent with a drought and an ENSO event. Temperatures reached 42.7 °C in Pretoria and 38.9 °C in Johannesburg on 6th January. | (Russo *et al* 2016, CRED 2020, Codjoe *et al* 2020, World Meteorological Organization 2016) |
| 2016 [MAM] | Sahel, Sahel-Sudano | Was the warmest April to date at the time in the Sahel. | | (Batté *et al* 2018) |
| 2016 [JJA] | Mediterranean, Sahara, Sahel, Sahel-Sudan and Guinea | | ENSO event | (World Meteorological Organization 2016) |
| 2016 [SON] | Mediterranean and Sahara | | ENSO event | (World Meteorological Organization 2016) |
| 2017 [MAM] | Mediterranean and Sahara | | On 17 May reported record of 42.9°C Larach station, northern Morocco | (World Meteorological Organization 2017) |
| 2018 [JJA] | Mediterranean and Sahara | | Algeria saw a peak of 51.3°C in July | (World Meteorological Organization 2018b) |
| 2019 [DJF] | Southern Africa | Reduction in avocado crops in some regions | | (Jansen 2019) |
| 2019 [MAM] | Sahel-Sudan | | In Nigeria 42.2°C recorded in Minna, 120km northwest of Abuja. Meanwhile, Kano, 345km north of the capital, has notched up highs in | (Al Jazeera 2019b) |



| | | | excess of 40°C every day since the beginning of April | |
|---|---|---|---|---|
| 2019 [JJA] | Mediterranean and Sahara | | Temperatures up to 47 °C in Morocco | (Morocco World News 2019) |
| 2019[SON] | Southern Africa | Court houses in Malawi allow for the wearing of wigs to be exempt. Increase in animal in particular elephant deaths in Botswana and Zimbabwe. One death reported in Botswana. | Temperatures up to 45°C in Malawi | (Sicetsha 2019, Al Jazeera 2019a, Tebele 2019) |
| 2020[JJA] | Mediterranean and Sahara | | Temperatures reached between 42 and 47°C in several provinces in southern Morocco | (Kasraoui 2020) |














## 3.2 South Africa 2015/16 Heatwave

The South Africa heatwave lasted from the 30th December 2015 to the 6th January 2016. It was chosen because it is one of the most widely reported heatwaves for the African continent (Table 1). In the 8 days (22nd to 29th December) leading up to the heatwave there are positive anomalies in the area and in much of the continent compared to the 1981 to 2010 climate for the mean of the UTCI, Z850 and Z500 (*figure 1,a*). However, there is a large negative anomaly in z500 and z850 off the coast of South Africa during this period, and the temperature is anomalously cooler than the climate average by up to -6°C.

During the 8 days of the heatwave all variables have positive anomalies in the area (*figure 1,b*).This indicates warmer than average temperatures and higher heat stress conditions coupled with anomalous high pressure. Much of the continent has positive anomalies in the UTCI, z850 and z500, whilst the temperature is slightly below average. In the 8 days (7th to 15th January) after the heatwave there is the return of the negative anomaly in the z500 and z800 off the coast of South Africa, extending onto the coast, and an area of low pressure (*figure 1,c*). Considering the temperature and the UTCI it can be inferred that this low pressure cools temperatures and heat stress from the heatwaves warm anomalies.

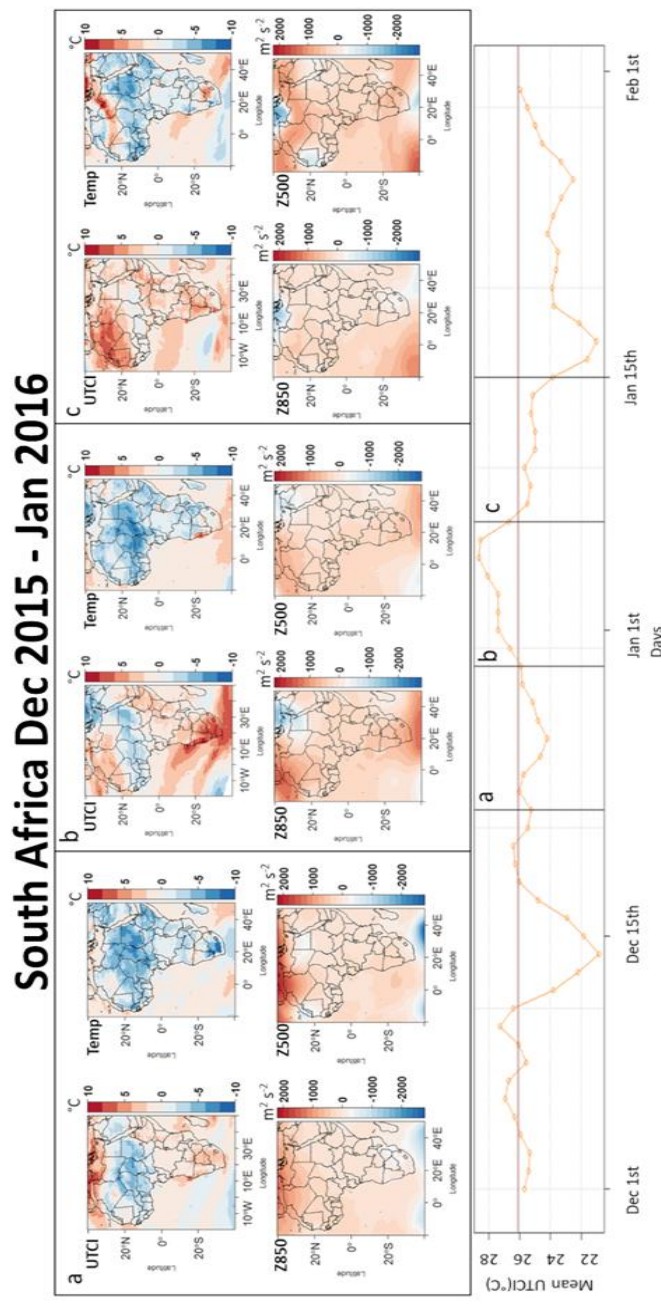


*Figure 1: Before, during and after the South Africa heatwave (2015/16 a: 22nd to 29th, b:*
*December, 30st to January 6th, c: 7th to 15th January), the Heatwave is indicated by when the*
*mean UTCI trend (orange) is above the 90th percentile 1981-2010 climatology (in red).*

### 3.3 Morocco 2000 Heatwave

The 2000 Morocco heatwave lasted from the 30[th] July to the 3[rd] August. It was chosen because it was the only heatwave with a reported economic impact in Table 1. In the 5 days (25[th] to the 29[th] July) leading up to the heatwave there are slight warm anomalies in the area compared to the 1981 to 2010 climate for both temperature and the UTCI (*figure 2,a*). There are also slight positive anomalies in geopotential at both z500 and z850. In comparison, the rest of the continent has slight cool anomalies in temperature and UTCI reaching -4*°C* and -6°C respectively.

During the 5 days of the heatwave there are significant warm anomalies in both the UTCI of up to 10°C and temperature of up to 6°C in Morocco (*figure 2, b).* In addition, within z500 and z850 there are positive anomalies in geopotential representative of the region being under the influence of high pressure. Interestingly the area of high warm anomalies is very small constrained to Morocco. In the 5 days (4[th] to the 9[th] July) after the heatwave the intensity of the warm anomaly for both temperature and UTCI dissipates and is at most 2°C and 1°C respectively (*figure 2, c)*. Within the z850 and z500 geopotential anomalies a negative anomaly can be seen to the east of Morocco, indicative of a low pressure system this could be in part having a cooling influence on Morocco.

Comparing the South Africa and Morocco heatwaves shows that the overall distribution of the peak in heat stress is similar with a rise in heat stress during the heatwaves of 2°C. However, this is where the similarities end, the Morocco Heatwave occurs over a shorter period (5 days) than that of the South African heatwave (8 days). In addition, the rise into the heatwave is steeper for Morocco increasing by 6°C in heat stress in 5 days than South Africa an increase of about 4°C in 8 days and this is mirrored at the end of the peak in heat stress. In addition, the area that a heatwave covers can be quite different, for example warm anomalies in the UTCI and temperature spread into neighbouring countries for South Africa, but are constrained to Morocco in July and August 2000. In both cases high pressure indicated by positive geopotential anomalies in the z500 and z850 dominates the area during the heatwave, with a low pressure system being in the vicinity as the heatwave dissipates.

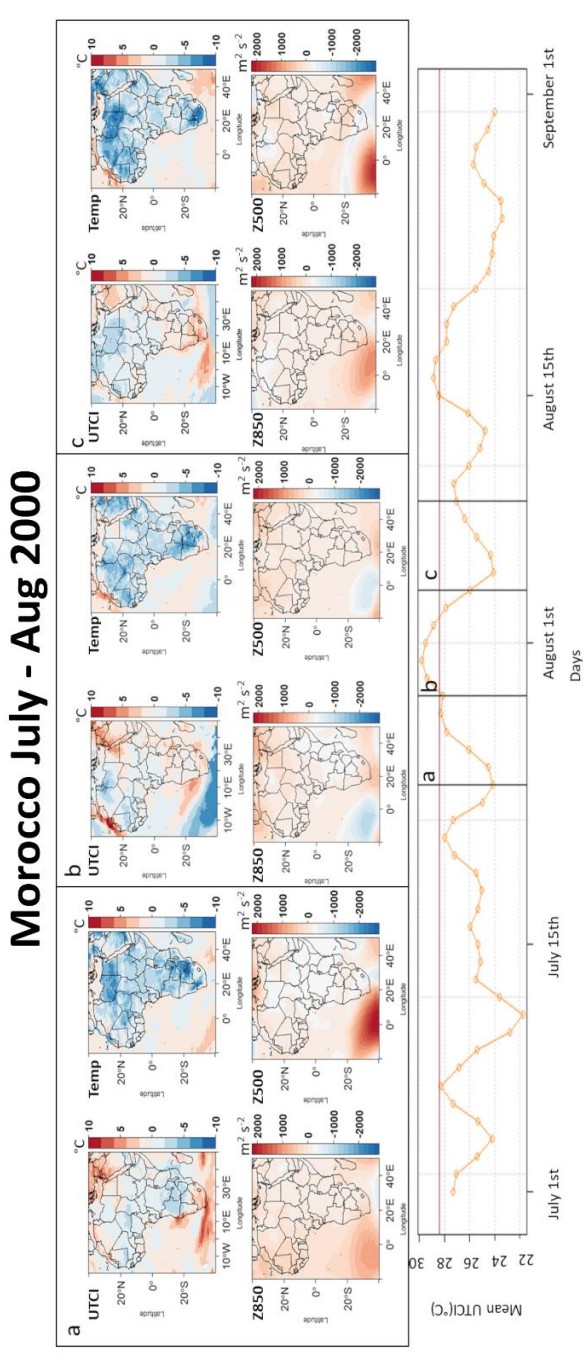


Figure 2: the Morocco (July-August 2000 a: 25th to 29th July, b: 30th July to 3rd August, c: 4th to 8th August). The heatwave is indicated by when the mean UTCI trend (orange) is above the 90th percentile 1981-2010 climatology (in red).


## 4. Discussion


The purpose of this study was to provide a list of heatwave case studies from literature for
Africa and then explore the characteristics of the pressure systems, temperature and
biothermal conditions of two of the heatwaves reported, to demonstrate the usefulness of
exploring past heatwaves. Further, our study supports others (i.e. Harrington and Otto, 2020;
van der Walt and Fitchett , 2021) suggesting that international databases such as EM-DAT
(CRED 2020) are not accurately recording heatwaves for Africa including sub-Saharan Africa
only including 7 out of our 39 listed heatwaves in their records, which is less than 20%.
In addition, more heatwaves are found for Africa by this study than is reported by any one
international meteorological organisation or disaster database (e.g. World Meteorological
Organisation and EM-DAT). Which supports previous findings of an existing discrepancy in
heatwaves reporting for Africa (Harrington and Otto 2020, van der Walt and Fitchett 2021).
However, there are some heatwave warning systems (Hafez and Almazroui 2016, Boubaker
2010) and there is some reporting in place for African Nations that is not always captured at
an international scale (Table 1).
With regards to the case studies, we show that anomalous levels of heat stress and
temperature have different extents of coverage for the South of Africa and Morocco
heatwaves. In addition, both heatwaves have positive geopotential anomalies at z500 and
z850 which is an indicative pressure pattern of heatwaves (Guigma *et al* 2020, Suarez-
Gutierrez *et al* 2020). This is the first time heatwaves from two different African regions have
been presented and compared using both their physical characteristics and reported impacts
and the results are as anticipated given that heatwaves drivers are closely linked to the
climatology and synoptic systems in a region (Russo *et al* 2016, Hu *et al* 2019).
We suggest it would be beneficial to investigate other heatwave events to further expand
this knowledge and reveal how heatwaves differ over the continent to inform the forecasting
and identification of heatwaves in African Nations. Future detailed analysis can also shed light
on the atmospheric dynamics of such extremes thus better aiding in-country forecasters in
the predictability of and preparedness to heatwaves (see e.g. Guigma et al. 2021).
Strengthening local meteorological services alongside health services has been identified as
a key element for climate change preparedness plans and public health policies addressing



temperature-related morbidity and mortality in Sub-Saharan Africa (Amegah *et al* 2016,
Nunfam *et al* 2019, Hussey and Arku 2020,van Loenhout *et al* 2021).
The reported impacts of the South of Africa heatwave were 11 deaths and 20 injures (*table*
*1*). Previous literature reports a 1.64% increase in excess mortality rates in South Africa above
an air temperature threshold of 19°C (Scovronick and Armstrong 2012). Considering that our
analysis shows positive anomalies in heat stress indicated by the UTCI of up to 10°C, we
hypothesise that impacts of the South Africa heatwaves have been underreported. For the
Morocco heatwave the reported impacts are 4 million chickens dying leading to up to 809,000
USD of damages (*table 1*). Interestingly, this is the only heatwave with an economic loss
associated with it. This reiterates the complexities of identify loss and damage due to heat
extremes, especially for Africa where research on heat-related impacts is still limited
(Campbell *et al* 2018).
Africa is exemplary in highlighting challenges that heatwave reporting faces globally. Whether
this is lack of observations, under-reporting fuelling a lack of evidence or a patchy research
field, these are intrinsic factors that has a growing mandate to be addressed globally
(Harrington and Otto 2020, Vicedo-Cabrera *et al* 2021). Other studies show that heatwaves
are increasing in frequency and intensity for Africa (Ceccherini *et al* 2017, Hu *et al* 2019, Russo
*et al* 2016) and it therefore should be a priority that the whole continent has access to robust
heat hazard forecasting.
**5. Conclusion**
Overall, this study presents a comprehensive list of reported heatwaves for Africa, which
includes more reported heatwaves than any other one source. In addition, this study supports
calls for more robust reporting of heatwaves in Africa. Further, characteristics including that
of the UTCI are explored for two of the reported heatwaves Morocco 2000 and South Africa
2015/16, this demonstrates that as might be anticipated heatwaves are more complex than
periods of high temperature and are linked to local synoptic systems and geography of a
region. This study recommends that other studies focus on the reported heatwaves to provide
evidence that heat is impacting African Nations. Finally, given the amount of reported
heatwaves and the changing climate the whole continent should have access to robust heat
hazard forecasting and resources for rigorous reporting.


## Code and Data Availability

ERA-5 Data is available freely from the Copernicus Data Store Website: DOI: 10.24381/cds.f17050d7

## Author contribution

**Chloe Brimicombe**: Conceptualization, Formal analysis, Investigation, Resources, Writing - original draft, Visualization.. **Claudia Di Napoli**: Conceptualization, Resources, Writing - review & editing. **Rosalind Cornforth:** Conceptualization, Writing - review & editing. **Florian Pappenberger:** Writing - review & editing. **Celia Petty**: Writing - review & editing. **Hannah L. Cloke**: Conceptualization, Supervision, Writing - review & editing.

## Declaration of Competing Interest

The authors declare that they have no known competing financial interests or personal relationships that could have appeared to influence the work reported in this paper.

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
