# Peer review of "Characteristics of Heatwaves in Africa: Morocco 2000 and South Africa 2015/16"

_Natural Hazards and Earth System Sciences, 2021_

## Referee Comment (RC1)

- **General comments**

The work here presented has the potential for being a relevant and useful study. The subject under discussion is of extreme importance considering the current climate change context, the natural and socio-economic impacts linked to heatwaves, the demographic projections for Africa and, finally, the lack of analysis conducted for this continent, where the living conditions of most of the population are poor and climate change is far from being a priority for the policymakers. I believe the authors share this perception and have in mind the relevance that studies of this kind have.

However, the manuscript calls for major revisions and needs to be improved in many aspects. First, regarding the quality of writing, in many sections the text is confused, disconnected and with grammatical and syntax errors. The reader often struggles to understand the rational and the ideas that authors pretend to share. Therefore, I strongly recommend a thorough review of the English writing throughout all the paper.

Some of the methods and metrics adopted by the authors must be clarified. For instance, the UTCI index lacks a mathematical definition and a more detailed interpretation of its values. I also have some doubts regarding some of the methodologies that were adopted by authors to achieve the objectives that were proposed. In addition, some results are not consistent with the interpretation and discussion conducted throughout the manuscript text. I found even some results "unexpected" and very strange, calling for an urgent review. As a consequence of this, most of the conclusions made by the authors find a weak support in the results presented.

The manuscript needs major reviews. I ask the authors to kindly consider the following comments and suggestions:

**Line 23**: What is "Morocco heat"?

**Line 26-27**: Please clarify this sentence… I'm not sure what authors are trying to say here.

**Line 32-33:** These deaths occurred mostly during two particular mega-heatwaves events in Europe. Authors should detail the particularities of these two episodes (duration, severity of the induced heat-stress levels) and explain why they were responsible for such high mortality rates and other impacts in many natural and socio-economic sectors… This would reinforce the importance of analyze in detail specific historical events such as the ones studied in this work.

**Lines 36 – 45:** This paragraph is very important to show the relevance of this study. It represents a substantial contribution to the understanding of natural hazards and their consequences. The authors should detail how the poor living conditions of most of the African population are crucial to explain the high levels of heat-related mortality and how the fragile economies struggle to recover from heatwaves and other climate extremes – showing once again the relevance of this type of studies, particularly for the African continent.

**Line 75:** Authors should provide a mathematical definition for the UTCI index. They should also provide a thorough interpretation of what this index really means. What are the variables involved and what are the implications for humans when extreme high/low values of this index are recorded?

**Line 93-97**: Check the syntax of this sentence.

**Line 98:** I'm not sure what authors mean by "notably area"; check the syntax. Also, try to use a more robust and objective reason (severity, magnitude of the events, spatial extend......) to justify the decision of choosing these two events in particular... I'm not convinced. Also, "Reported Impacts"? Which ones?

**Line 118-119**: Poor English writing.

**Line 122**: Showing the coordinates of these regions only makes sense if the authors reference a figure....

**Line 125-126**: I'm confused. This is not clear. Rewrite this sentence please.

**Figure 1 and Figure 2:**

- The caption of the figure is very poor. Authors should provide more information about the variables and the figures that are presented: What are the panels showing? Anomalies? "Temp" is surface temperature?
- The location of South Africa (Morocco) within the African continent should be highlighted.
- I don't like the layout (horizontal) of the figure... It's not easy for the reader to follow the figure and the text at the same time. Try to order the panels in a vertical way.
- Try to save some space by removing the colorbars and the legends of the x-axis/y-axis that are repeated throughout the several panels.
- The bottom panel with the UTCI time series is supposed to be Figure 1d, isn't it? Also, are these area averaged UTCI values for South Africa (Morocco in Figure 2)? This is not mentioned... Please clarify.

**Line 157**: I'm not sure about the need to show the Z850 anomalies considering that you are already showing the 500-hpa geopotential anomaly. If your idea is to prove that these heatwaves are linked to the establishment of a quasi-stationary anticyclonic system, the 500-hpa captures these structures well. Typically, in this type of analysis performed for mid-latitude regions, the 500hpa atmospheric level is enough to find these synoptic systems. In addition, it seems that authors are not extracting any relevant information from z850 field. My suggestion would be to go to a single panel where the 500-hpa geopotential anomaly would be represented by contours and the 850-hpa temperature anomaly by colors. The temperature at this level (just above the boundary layer where the diurnal cycle is almost negligible) would give you information about potential warm air masses that are being advected to the regions of interest and that could be heated due to subsidence. This would also allow the authors to save a panel on an already heavy figure.

**Line 160 – 161**: Is this true for Temp? Over South Africa, the anomalies are close to zero! I have doubts about the quality of these results. If the temperature values are within the expected values, we can't consider this as a Heatwave. For instance, Pretoria region has even negatives anomalies of Temp (once again, what is Temp??)... In table 1 authors say the following:

"*Heatwave Concurrent with a drought and an ENSO event. Temperatures reached 42.7 °C in Pretoria and 38.9 °C in Johannesburg on 6 th January*." Something here is not right. I know that the heatwave definition used by authors is based on the UTCI values, but, isn't it strange to have a Heatwave event with lower temperatures than the expected values? Once again, how are the UTCI values obtained? This needs a careful and urgent review!!

**Line 165:** It should be Z850... Once again, the authors are not extracting any relevant information from the Z850 anomaly field.

**Line 166 – 167:** Not sure about what the authors mean to say here. Please clarify

**Line 190**: Slight cool anomalies? According to the results, there are regions reaching values between -5ºC and -10ºC.

**Line 196 – 198**: Not sure about this... Morocco is still under the influence of high anomalies of Temp (reaching levels very close to 5ºC). Authors need to Check the data because the results are not consistent with what is discussed in the text.

**Line 198 – 200:** Please rewrite this sentence.

**Lines 200 – 201**: Please rewrite this sentence. Not sure about what authors are trying to say here.

**Lines 232 – 234**: Authors should include some possible explanations for this, for instance, the topography, the influence of more local meteorological processes, the influence of the Sahara Desert, different land covers….

**Lines 234 – 236**: Which pressure pattern? Authors were able to link heatwaves in South Africa and Morocco with geopotential anomalies. Here, in the discussion, they should detail what are the synoptic processes that are behind this link.

**Lines 236 – 239**: Authors should rewrite this sentence please.

**Line 240**: This would be the ideal paragraph to authors mention some of the limitations of this study.

**Lines 259 – 260:** Authors should rewrite this sentence please.

**Lines 267 – 276**. Authors should rewrite all the Conclusion paragraph. There are a lot of syntax and grammatical errors. The text is confused and disconnected. I'm not sure if authors were successful in covering the ideal bullets for the Conclusion section (Why is this study relevant? What were the key findings and what can we learn from them? Is this a novel approach for Africa? How could these results be used to predict future events? Should policymakers and authorities start to look at these climatic extreme with more attention? Basically, why this work should be publish and shared.